# Understanding the Dosage-Dependent Role of *Dicer1* in Thyroid Tumorigenesis

**DOI:** 10.3390/ijms251910701

**Published:** 2024-10-04

**Authors:** María Rojo-Pardillo, Ludivine Godefroid, Geneviève Dom, Anne Lefort, Frederick Libert, Bernard Robaye, Carine Maenhaut

**Affiliations:** 1IRIBHM J. E. Dumont, Université Libre de Bruxelles, 1070 Brussels, Belgium; maria.rojo.pardillo@ulb.be (M.R.-P.);; 2BRIGHTcore Facility, 1070 Brussels, Belgium; 3IRIBHM J. E. Dumont, Université Libre de Bruxelles, Campus Charleroi, 6041 Charleroi, Belgium

**Keywords:** *Dicer1*, microRNAs, papillary thyroid carcinoma, thyroid cancer, thyroid

## Abstract

Tumors originating from thyroid follicular cells are the most common endocrine tumors, with rising incidence. Despite a generally good prognosis, up to 20% of patients experience recurrence and persistence, highlighting the need to identify the underlying molecular mechanisms. *Dicer1* has been found to be altered in papillary thyroid cancer (PTC). Studies suggest that *Dicer1* functions as a haploinsufficient tumor suppressor gene: partial loss promotes tumorigenesis, while complete loss prevents it. To investigate the effects of partial or total *Dicer1* loss in PTC in vitro, we generated stable *Dicer1* (+/−) cell lines from TPC1 using CRISPR-Cas9, though no *Dicer1* (−/−) lines could be produced. Therefore, siRNA against *Dicer1* was transfected into *Dicer1* (+/−) cell lines to further decrease its expression. Transcriptomic analysis revealed changes in proliferation and cell locomotion. BrdU staining indicated a slow-down of the cell cycle, with fewer cells in S phase and more in G0-G1-phase. Additionally, transwell assays showed decreased invasion and migration after *Dicer1* knockdown by siRNA. Moreover, *Dicer1* overexpression led to decreased proliferation, invasion, and increased apoptosis. Our findings deepen the understanding of *Dicer1*’s role in thyroid cancer, demonstrating that both complete elimination and overexpression of *Dicer1* inhibit thyroid oncogenesis, highlighting *Dicer1* as a promising target for novel therapeutic strategies.

## 1. Introduction

Tumors originating from thyroid follicular cells are the most frequent endocrine tumors, with an increasing incidence that is projected to make thyroid cancer one of the most common cancers in women within the next decade, particularly affecting women between the ages of 30 and 50 [1,2]. Chemotherapy, radiotherapy, and surgical interventions remain the primary treatment options. However, one of the significant challenges in treating thyroid cancer is the development of resistance to standard treatments, which can lead to disease progression, reduced treatment efficacy, and higher mortality rates, posing a significant challenge to patient outcomes [3]. Therefore, studying thyroid cancer is crucial, not only because of its rising prevalence but also due to its significant impact on patient quality of life and long-term health outcomes. Additionally, there is an acute need to identify the molecular mechanisms underlying benign and aggressive clinicopathological features, as well as to explore novel therapeutic targets and follow-up strategies.

Thyroid cancers originate from two distinct types of endocrine thyroid cells: follicular thyroid cells and parafollicular C cells. The 2022 WHO classification categorizes neoplasms derived from follicular cells into three groups: benign neoplasms, low-risk neoplasms, and malignant neoplasms [2]. Malignant neoplasms comprise papillary thyroid cancer (PTC), follicular thyroid cancer (FTC), poorly differentiated thyroid cancer (PDTC), and undifferentiated (anaplastic) thyroid cancer (ATC). Papillary thyroid cancer stands as the predominant form of thyroid cancer, representing up to ~85% of thyroid malignancies [4]. Despite a good prognosis, up to 20% of PTC patients are facing persistence and recurrence [5]. The initiation and progression of PTC are significantly influenced by genetic alterations within the mitogen-associated protein kinase (MAPK) pathway such as BRAF and RAS point mutations or RET/PTC rearrangements [6]. Anaplastic thyroid carcinoma can arise primarily de novo or from pre-existing PTC or FTC, and currently, there are no effective treatments available.

MicroRNAs (miRNAs) are RNA molecules of approximately 22 nucleotides in length that play a crucial role in post-transcriptional gene regulation via binding to target messenger RNAs (mRNAs), thereby modulating their stability and translation efficiency [7,8]. Most miRNAs are generated by a canonical pathway in which the type III ribonuclease *Dicer1* cleaves miRNA precursors to form mature miRNA duplexes. Mature miRNAs are then incorporated into the RNA-induced silencing complex (RISC), guiding by base-pairing the complex to negatively regulate target mRNA expression. MiRNAs are involved in the regulation of a variety of biological processes, such as cell cycle, differentiation, proliferation, apoptosis, stress tolerance, energy metabolism, and immune response [3], and their deregulation is linked to multiple pathologies such as cancer, among which is thyroid cancer [9,10,11,12,13,14,15]. Given that a single anomaly in the components of the miRNA biogenesis pathway has the potential to impact the entirety of miRNA production, several studies hypothesized that altered expression of miRNA processing enzymes could be the cause of the dysregulation of microRNAs.

Heterozygous mutations in *Dicer1* are recurrent in diverse cancers [16,17,18,19]. *Dicer1* syndrome is a rare genetic condition caused by germline mutations in the *Dicer1* gene and predisposes to hereditary cancers such as pleuropulmonary blastoma, cystic nephroma, or differentiated thyroid carcinoma, among others [20]. While germline mutations are distributed across the 27 exons of *Dicer1*, somatic mutations are predominantly found in exons 24 and 25, which encode the two catalytic domains (RNase IIIa and RNase IIIb) [21]. Interestingly, loss-of-heterozygosity (LOH) is rare among *Dicer1*-mutation-associated diseases [22]. Beside *Dicer1* mutations, lower *Dicer1* expression has been observed in numerous cancers and has been associated with aggressive features in thyroid cancer [23]. 

Several studies reported that while monoallelic inactivation of *Dicer1* promotes tumorigenesis, biallelic loss inhibits tumorigenesis [24,25]. Furthermore, in vivo mice models showed that tumors originating from *Dicer1* (floxed/floxed) animals consistently maintained at least one functional *Dicer1* allele, indicating a selection against the complete loss of *Dicer1* expression and enforcing the idea that *Dicer1* complete loss inhibits tumor development [21,25]. Considering all these results, *Dicer1* has been proposed as a haploinsufficient tumor suppressor, but the biological consequences of *Dicer1* partial or complete elimination differ is based on the specific organ targeted, indicating that the role of *Dicer1* may not exclusively depend on dosage, but also on tissue context. 

In this study, we aimed to better understand the response of thyroid cancer cells to partial or total loss of *Dicer1* in vitro. Heterozygous *Dicer1* (+/−) cell lines were established via CRISPR/Cas9. As we were unable to generate homozygous *Dicer1* (−/−) cell lines, cells were transfected with *Dicer1* siRNA to further reduce *Dicer1* mRNA and protein expression. Our results suggest that *Dicer1* exerts a dosage-dependent effect in thyroid cell lines.

## 2. Results

### 2.1. Dicer1 mRNA Expression Is Downregulated in Human Papillary Thyroid Carcinoma (PTC) and in the PTC Cancer-Derived Cell Line TPC1

Analysis of Dicer1 mRNA expression in normal tissues (n = 59), thyroid tumors (PTC, n = 505), and metastatic samples (n = 8), using data from The Cancer Genome Atlas (TCGA), revealed significant downregulation of Dicer1 mRNA in PTC and metastases compared with normal thyroid tissues (Figure 1A). Further investigation through quantitative RT-PCR analysis in nine independent PTCs and their adjacent normal tissues confirmed the downregulation of Dicer1 mRNA (Figure 1B). Additionally, Dicer1 expression was evaluated in the PTC-derived cell line TPC1. A significant decrease in Dicer1 mRNA expression was observed in TPC1 cells compared with a pool of normal human thyroid tissues (Figure 1C). Given the important role of Dicer1 in miRNA biogenesis and the known role of miRNA in thyroid tumorigenesis, these observations suggested a potential role for this protein in thyroid cancer.

### 2.2. Dicer1 (+/−) Cell Lines Were Generated from TPC1 Using CRISPR-Cas9 and Showed a 50% Decrease in Dicer1 Protein Expression

TPC1 cells were transfected with a CRISPR-Cas9 plasmid containing the guide sequences against *Dicer1* in order to generate *Dicer1* (+/−) or (−/−) cell lines. Following transfection, cells were sorted based on mCherry fluorescence and plated individually to obtain monoclonal populations. Up to ~100 clones were isolated and analyzed using Sanger sequencing. Three *Dicer1* (+/−) clones, designated as D11, WB1, and TD5, were identified and selected by comparison to the wild-type unmodified sequence, showing a loss of 20, 7, and 185 nucleotides, respectively (Figure 2A), leading to a frameshift resulting in the creation of a premature stop codon truncating the protein. These resultant cell lines retained one wild-type *Dicer1* allele (+). Further assessment of Dicer1 protein expression via Western blot analysis, with an antibody targeting the C-terminal region of the protein, confirmed a significant reduction (~50%) in Dicer1 protein levels in the generated *Dicer1* (+/−) cell lines compared with wild-type TPC1 cells (Figure 2B,C).

In the aim of isolating *Dicer1* (−/−) cell lines, we performed a second CRISPR-Cas9 experiment by transfecting our CRISPR-Cas9 plasmid into the newly established *Dicer1* heterozygous cell lines. However, despite multiple attempts, we were unable to isolate any clones lacking both *Dicer1* alleles, suggesting that homozygous deletion of *Dicer1* is lethal. This observation is consistent with previous mutational analyses in humans indicating that while partial loss of *Dicer1* is linked to numerous cancers, complete loss is rarely observed and poorly tolerated [23].

### 2.3. Heterozygous Loss of Dicer1 Does Not Induce Major Changes in TPC1 Cells’ Behavior

First, we carried out an RNA sequencing experiment to assess the effects of the heterozygous loss of Dicer1 on the global transcriptome, comparing parental TPC1 cell line with the three Dicer1 (+/−) cell lines individually (Gene Expression Omnibus, GSE273204). Although the Principal Analysis Component (PCA) analysis revealed changes between the different cell lines, the global variance was low (Figure 3A) and very few genes were identified as commonly upregulated (85 genes, Figure 3B) or downregulated (52 genes, Figure 3C) in all three heterozygous cell lines. The expression of nine of these genes was quantified via RT-qPCR and the results confirmed the RNAseq observations (Figure 3D). Further analysis of the RNAseq data did not reveal any major pathway commonly upregulated or downregulated.

Then, to investigate the impact of Dicer1 heterozygous deletion on cell behavior, we performed proliferation (Figure 4A), apoptosis (Figure 4B), and migration or invasion (Figure 4C,D) analyses. No differences were observed between the parental TPC1 cell line and the heterozygous Dicer1 (+/−) cell lines D11, WB1, and TD5, supporting the previous results obtained by the RNAseq, where no major pathways were observed deregulated.

### 2.4. Dicer1 mRNA and Protein Expression Are Further Reduced by Dicer1 siRNA Transfection

To further reduce Dicer1 expression, TPC1 cells were transfected with anti-Dicer1 siRNA. Transfection efficiency was assessed through flow cytometry using a fluorescent mimic SIGLO (5 mM) and was confirmed to be greater than 98% across all the cell lines studied (Appendix A). Additionally, we evaluated the stability of the transfected siRNA for 6 days following transfection. Dicer1 was undetectable from days 3 to 6 (Appendix A), followed by re-expression starting on day 7. Based on these results, we decided to focus on day 4 post-transfection as our primary time point for analysis, and all subsequent data presented in this study are based on measurements obtained at this time. 

We then evaluated Dicer1 mRNA expression levels in the TPC1, D11, WB1, and TD5 cell lines and observed a strong decrease in mRNA expression in cells transfected with siRNA against Dicer1 compared with the control group (Figure 5A). Western blot analysis of Dicer1 protein expression following Dicer1 siRNA transfection revealed undetectable levels of the Dicer1 protein (Figure 5B,C). Given these data, confirming a drastic decrease in Dicer1 mRNA and an undetectable protein level following Dicer1 siRNA transfection, we considered this model a total Dicer1 defective model, allowing us to address the functional and molecular implications of complete Dicer1 loss in the context of thyroid cancer.

### 2.5. RNA Sequencing Reveals Transcriptomic Alterations Impacting Cell Proliferation and Cell Motility in Cells Transfected with Dicer1 siRNA

We carried out a second RNAseq experiment (Gene Expression Omnibus, GSE273129) to assess the effects of total loss of Dicer1 following Dicer1 siRNA transfection compared to the control group in two cell lines, TPC1 and D11. The control group was composed of non-transfected cells and cells transfected with the negative control. Indeed, no differences were found between these two experimental conditions in the RNAseq results. 

In D11 cells, 1123 upregulated and 1204 downregulated genes were significantly differentially expressed (FDR 0.05 and minimum fold change 1.5×) in Dicer1 siRNA transfected cells compared with the control samples (Figure 6A). Further analysis using GOrilla Biological function dataset revealed the top 10 upregulated (Figure 6B) and downregulated (Figure 6C) pathways in these cells. These results and pathways were similar in TPC1 cells transfected with Dicer1 siRNA. We observed two main axes that were deregulated; on one hand, an upregulation of pathways involved in cell movement such as cell motility, cell migration, locomotion, etc., and on the other hand, a downregulation of multiple pathways linked to proliferation such as regulation of the cell cycle process, cell cycle phase transition, mitotic cell cycle process, cell division, etc. Functional assays exploring these major pathways were then performed to validate and further understand the biological processes highlighted through the gene expression data.

### 2.6. Complete Depletion of Dicer1 Inhibits Proliferation and Delays Progression through the Cell Cycle

To evaluate the impact of Dicer1 loss on cell proliferation, we conducted cell counting four days after transfection of Dicer1 siRNA in TPC1 and in the Dicer1 (+/−) cell lines. Dicer1 defective cells exhibited a 30 to 40% reduction in cell count compared with the negative control (Figure 7A). To further study the proliferation capacity of the cells, we performed BrdU-7-AAD intracellular staining and flow cytometry analysis. Bromodeoxyuridine (BrdU) is a synthetic nucleoside analog of thymidine, which is incorporated into DNA during active DNA synthesis while 7-aminoactinomycin D (7-AAD) is a fluorescent dye that binds to DNA through intercalating between the base pairs, commonly used to differentiate cells based on their DNA content. Cells that are actively proliferating (in S phase) will incorporate BrdU into their newly synthesized DNA. Cells with higher DNA content (in G2 and M phases) will show higher concentrations of 7-AAD than cells with lower DNA content (in G0 and G1 phases).

Our results revealed a significant decrease (≃35%) in the percentage of cells in S phase in cells transfected with Dicer1 siRNA (siDICER) compared with cells transfected with the negative control (NC), in all four cell lines (Figure 7B,C). These results were confirmed via EdU intracellular staining (Appendix A). Furthermore, we observed an increase of 20 to 35% of the number of cells in G0/G1 phases following Dicer1 siRNA transfection (siDICER) (Figure 7B,D). No changes were observed regarding the number of cells in G2/M phases (Appendix A). 

These observations were in accordance with our RNAseq analysis (Figure 6). Notably, we observed a downregulation of cyclins A, B, and E, while cyclin D1 expression remained unchanged. The protein expression of cyclin D1 and cyclin E, which control the entry into the S phase of the cell cycle, was verified via Western blotting and confirmed the RNAseq observations (Figure 8). To further analyze the regulators of the cell cycle, we focused our attention on different cell cycle inhibitors: p16 (Cyclin Dependent Kinase Inhibitor 2A, CDKN2A), p21 (Cyclin Dependent Kinase Inhibitor 1A, CDKN1A), and p27 (Cyclin Dependent Kinase Inhibitor 1B, CDKN1B). Their expressions did not show any modulation in our RNAseq analysis and these results were confirmed via RT-qPCR (Appendix A).

### 2.7. Total Loss of Dicer1 Leads to a Rise in Apoptosis

*Dicer1* siRNA transfection led to a significant increase in caspase 3/7 activity (Figure 9A), suggesting the involvement of Dicer1 in the regulation of apoptotic pathways. In addition, Annexin V/PI analysis demonstrated a significant increase in apoptosis following Dicer1 knockdown by siRNA (Figure 9B and Appendix A)). These results are consistent with our previous findings using the caspase 3/7 activity assay, reinforcing the validity of our data. Thus, our results robustly support the induction of apoptosis in cells lacking Dicer1.

### 2.8. Total Loss of Dicer1 Inhibits Migration and Invasion

Our RNAseq analysis highlighted an alteration of pathways involved in cell movement, hence migration and invasion were assessed using transwell migration or invasion chambers. Downregulation of Dicer1 via siRNA transfection strongly reduced the number of migrating (Figure 10A,C) and invasive (Figure 10B,D) cells.

### 2.9. Dicer1 Knockdown Exerts Similar Effects in H-Tori3 and BCPAP Cells

To see if our observations can be generalized, we investigated two other thyroid cell lines: BCPAP, derived from a poorly differentiated thyroid carcinoma, and H-Tori3, derived from normal thyrocytes immortalized by SV40. 

Both showed a significant decrease in Dicer1 mRNA expression compared with a pool of normal human thyroid tissues (Figure 11A). First, we analyzed transfection efficiency in BCPAP and H-Tori3 cell lines through flow cytometry using a fluorescent mimic SIGLO (5 mM), and transfection efficiency was confirmed to be greater than 98% in both cell lines (Appendix A). Then, we confirmed that Dicer1 protein expression was significantly diminished following Dicer1 siRNA transfection, reaching values close to zero (Figure 11B). As previously observed in TPC1 and TPC1-derived Dicer1 (+/−) cell lines (Figure 7, Figure 9 and Figure 10), our findings demonstrated a notable alteration of cell cycle progress, namely a reduction in the percentage of cells in S phase (Figure 11C) and increase in the number of cells in the G0/G1 phase (Figure 11D), though we did not observe significant changes in the amount of cells in the G2/M phase (Appendix A). 

Finally, we examined the invasion and migration capacities of H-Tori3 and BCPAP cells four days post-transfection with Dicer1 siRNA and obtained results similar to those obtained with the TPC1- and Dicer1 (+/−) TPC1-derived cell lines: Dicer1 siRNA transfection significantly decreased the number of migrating (Figure 11E) and invasive cells (Figure 11F). 

So, the effects of Dicer1 deletion on proliferation, migration, and invasive properties of TPC1 are not restricted to this particular tumor cell line, as they were reproduced in another tumor cell line (BCPAP) as well as in a non-tumoral cell line (H-Tori3). 

### 2.10. Increasing Dicer1 Levels Led to a Decrease in Proliferation and Invasion While Rising Apoptosis 

Reasoning that TPC1 and the Dicer1 (+/−) TPC1 cell lines might both be considered as having decreased Dicer1 expression compared with normal thyroid cells (Figure 1C), even if at different levels, we decided to investigate whether increasing Dicer1 expression could modify cell behavior. Dicer1 mRNA was highly expressed for four days following Dicer1 plasmid transfection (Appendix A). Our findings revealed that two days post-transfection, increasing Dicer1 expression in TPC1 cells led to a reduction in the number of proliferating cells (Figure 12A) and a rise in apoptosis (Figure 12B,C). Invasion rates were also reduced (Figure 12D). 

## 3. Discussion

The 2022 WHO classification distinguishes four malignant neoplasms derived from follicular cells: papillary thyroid carcinoma (PTC), follicular thyroid carcinoma (FTC), poorly differentiated thyroid carcinoma (PDTC), and anaplastic thyroid carcinoma (ATC) [2]. While PTC and FTC are still partly differentiated tumors and have a good prognosis, ATCs are totally dedifferentiated tumors and do not respond to any therapy. Aberrant expression of miRNAs is often correlated to tumor development [9,10,11,12,13,14,15]. Whereas PTC and FTC present up- and downregulated miRNAs, ATC show mostly downregulated miRNAs [26,27], suggesting that miRNA might play a role in the transition from slow progressing differentiated papillary or follicular carcinoma into aggressive PDTC and ATC. This loss of miRNA may result from alterations in the miRNA biogenesis pathway. 

In line with this, downregulating or inactivating mutations of *Dicer1*, a type III ribonuclease involved in miRNA biogenesis has been described in thyroid and other cancers [16,17,18,19,28]. *Dicer1* syndrome is an autosomal dominant tumor predisposition syndrome caused by germline mutations in *Dicer1* [29]. *Dicer1* mutations, though rare, have been found in certain thyroid cancers, including the more aggressive variants [30,31]. Furthermore, *Dicer1* mRNA is downregulated in malignant thyroid samples and cell lines compared with normal tissues, and this downregulation is associated with aggressive features [23]. We confirmed this downregulation in nine PTCs and their adjacent tissues and showed that the thyroid cancer-derived cell lines TPC1 and BCPAP present very low levels of *Dicer1*. However, despite the fact that reduced levels of *Dicer1* mRNA have been reported in tumors, a loss of heterozygosity in *Dicer1* is rare [21]. 

*Dicer1* has been suggested to be a haploinsufficient tumor suppressor gene [24]. Previous in vivo studies in retinoblastoma, prostate, or lung cancer have reported that the loss of a single allele of *Dicer1* enhances tumorigenesis, while loss of both inhibits this process [25,32,33]. So far, partial or total depletion of *Dicer1* has never been addressed in a thyroid tumoral context. 

In order to study the role of *Dicer1* in thyroid cancer, we carried out heterozygous *Dicer1* deletion in the TPC1 cell line via CRISPR-Cas9 and we were able to generate three *Dicer1* (+/−) cell lines with partial inactivation of *Dicer1*. To assess the global effect of the heterozygous loss of *Dicer1* on the transcriptome, we performed RNA sequencing of the three *Dicer1* (+/−) cell lines, but no major pathways were observed to be deregulated. Although deregulated genes were found, these genes do not appear to be associated with any specific process. These observations were confirmed via proliferation, apoptosis, migration and invasion experiments, where no differences were observed between wild-type TPC1 and *Dicer1* (+/−) TPC1 cells. 

Despite multiple attempts, we failed to generate cell lines carrying homozygous deletion of *Dicer1*, supporting the idea that *Dicer1* loss may be detrimental or lethal [21,22]. Hence, to further reduce *Dicer1* levels in vitro we performed transfections with anti-*Dicer1* siRNA to knockdown *Dicer1* mRNA and protein expression. In these conditions, Dicer1 protein levels were undetectable and we considered these cells as total *Dicer1* defective cells. RNAseq analysis showed alterations in proliferation and cell locomotion-related pathways four days after *Dicer1* siRNA transfection. Further analyses revealed that *Dicer1* silencing led to reduced proliferation, delaying the G1/S transition through decreased *cyclin E* levels, reduced migration and invasion, and increased apoptosis levels. 

Our findings can be generalized as they were confirmed in two supplementary thyroid cell lines, H-Tori3 and BCPAP. Also, they are consistent with previous studies in PCCL3 rat thyroid cells and in murine sarcoma cell lines showing decreased proliferation and higher levels of apoptosis following *Dicer1* silencing [34,35]. On the other hand, Ramirez-Moya observed increased proliferation, migration, and invasion in TPC1 following *Dicer1* downregulation [28]. This discrepancy between our results and those of Ramirez-Moya’s could be explained through the different experimental conditions in the two studies: we conducted our experiments four days after *Dicer1* siRNA transfection, and at that time Dicer1 protein levels were undetectable, while their experiments were performed 48 h after *Dicer1* siRNA transfection, when low levels of Dicer1 protein were still detected via Western blotting. Consequently, these results seem discordant but are not if we consider Swahari’s hypothesis [24], suggesting that intermediate *Dicer1* levels promote tumor progression, possibly leading to enhanced proliferation, invasion, and migration, while complete loss of *Dicer1* inhibits tumorigenesis and results in cell death. 

Considering our results, it appears clearly that total loss of *Dicer1* in thyroid papillary carcinoma cell lines is deleterious to cell proliferation, migration, and invasion, and by extension to tumor development. However, we did not find any impact of heterozygous loss of *Dicer1* in TPC1. We believe that the fact that the TPC1 cell line itself presents a downregulation of *Dicer1* compared with normal thyroid cells may be the cause. TPC1 cells already show low *Dicer1* expression and even if *Dicer1* levels are further decreased in the heterozygous TPC1 *Dicer1* (+/−) cell lines, all those cells have decreased expression of *Dicer1* compared with normal thyrocytes, with similar effects on proliferation, apoptosis, migration, and invasion. So, to reach higher levels of the protein, as encountered in normal thyroid cells, *Dicer1* plasmid transfection was performed and resulted in a reduction in proliferation and invasion, as well as an increase in apoptosis. Our migration studies were not conclusive. Future studies will need to incorporate more advanced migration assays, such as wound healing assays or live cell imaging, as well as new time points after plasmid transfection, to provide a more comprehensive understanding of the mechanisms involved in migration. Figure 13 summarizes the dosage-dependent impact of *Dicer1* on thyroid cell behavior (Figure 13A) and the procedures that were used in this study to modulate *Dicer1* expression in TPC1 cells (Figure 13B). 

Besides the role of *Dicer1* in miRNA biogenesis, a miRNA-independent function based on its role in DNA damage repair has been proposed [36,37]. DNA damage response RNA (ddRNA), produced by the enzymes *Dicer1* and Drosha, are crucial for the efficient repair of double-strand breaks (DSBs) in DNA. While *Dicer1* WT tumors can efficiently repair DNA damage, *Dicer1* KO tumors can accumulate extensive DNA damage, leading to cell death. *Dicer1* heterozygous tumors can show partial DNA damage response and therefore increased mutagenesis frequency, promoting cancer progression. Nevertheless, in our in vitro model, we did not observe histone H2AX phosphorylation due to partial or total loss of *Dicer1* (Appendix A), suggesting the absence of DNA lesions. Furthermore, *Dicer1* has been reported to play a role in senescence. Mudhasani observed that *Dicer1* ablation led to upregulated p16 and p53 levels and induced a premature senescence in embryonic fibroblasts [38]. We did not observe any differences in p16 or p21 mRNA expression after *Dicer1* siRNA transfection in TPC1 and TPC1 *Dicer1* (+/−)-derived cells. Additional analysis of senescence-associated β-galactosidase (SA-β-gal) staining will be conducted to further investigate this biological process.

## 4. Materials and Methods

### 4.1. Human Tissue Samples 

Nine frozen PTC samples along with their adjacent normal tissues were obtained from the Jules Bordet Institute (Brussels, Belgium). This study was conducted in accordance with the Declaration of Helsinki and approved by the Ethics Committee of the Institut J.Bordet (Brussels, Belgium, protocol 1978-01/12/2016). Hematoxylin and eosin staining was performed on all samples, and their pathological status was verified by an anatomopathologist from the J. Bordet Institute. The protocols used in this study were approved by the ethics committee of J. Bordet Institute, and written informed consent was obtained from all participants.

#### In Silico Analysis

Human tissue sample data for in silico analyses were obtained from The Cancer Genome Atlas (TCGA) through the UCSC Xena Browser (https://xenabrowser.net/), accessed on 1 February 2024. Data from TCGA for thyroid cancer (THCA) were analyzed with *Dicer1* gene expression as the primary variable and phenotypic sample type (primary tumor, solid normal tissue, and metastatic samples) as the secondary variable. 

### 4.2. RNA Extraction and Quantitative PCR Amplification

Total RNA from human samples or cells was extracted using a miRNeasy mini kit (2170004, Qiagen, Antwerp, Belgium) according to the manufacturer’s instructions. The total amount of RNA was quantified and treated with DNAse I (18068015, ThermoFisher Scientific, Dilbeek, Belgium) and RNase Out (100000840, ThermoFisher Scientific). Reverse transcription was performed using a Superscript II reverse transcription kit (18064022, ThermoFisher Scientific). Quantitative PCR amplification was performed using KAPA SYBR FAST (KK4601, KapaBiosystems, Wilmington, MA, USA). The qPCR primers are listed in Appendix A. *NEDD8* and *TTC1* were used as internal normalizers [39]. The expression of each gene was calculated and normalized using the 2^−∆∆Ct^ method [40].

### 4.3. Cell Lines

The STR profile of TPC1, H-Tori3, and BCPAP cell lines was performed to ensure their purity and identity. The three cell lines were maintained in RPMI medium (52400025, ThermoFisher Scientific) supplemented with 10% fetal bovine serum, 2% penicillin/streptomycin, and 1% amphotericin B and cultured at a constant temperature of 37 °C in a humidified atmosphere containing 5% CO_2_. Heterozygous *Dicer1* (+/−) cell lines were established through the use of a CRISPR/Cas9 plasmid (99154, Addgene, Watertown, MA, USA) transiently transfected into TPC1 with Lipofectamine 3000 (L3000001, ThermoFisher Scientific) according to the manufacturer’s instructions. Target single guided RNA (sgRNA) sequences were identified with an online CRISPR design software (CHOPCHOP, version 3). The guide RNA (gRNA) sequence chosen was 5′GCCGTGTTGATTGTGACTCGTGG3′ located in exon 11 of *Dicer1*. Cells were collected 24–48 h after transfection, analyzed via flow cytometry and plated into 96-well plates with 1 cell per well. Colonies were expanded for cryopreservation, protein extraction, and genomic DNA extraction. 

Genomic DNA extraction was performed using a DNeasy Blood & Tissue Kit (69504, Qiagen). Genomic region surrounding the CRISPR target site for the *Dicer1* gene was amplified via PCR (Forward: 5′TGTAGGTACAGAGGCAGACAG3′ and Reverse: 5′CTCTCTTGGTGTCTGGGTGA3′). The resulting PCR products were cloned into a plasmid using Zero Blunt™ TOPO™ PCR Cloning Kit (K280020, Thermo Fisher) for sequencing to verify gene disruption.

### 4.4. Cell Transfection

Transfection efficiency was assessed via flow cytometry using a fluorescent mimic SIGLO (5 nM) (D-001630-01-05, Dharmacon, Lafayette, CO, USA). Cells were transiently transfected with 5 nM of silencer select *Dicer1* siRNA (s23754, ThermoFisher Scientific) or negative control (4390846, ThermoFisher Scientific). The siRNA transfection was performed using Lipofectamine RNAiMAX reagent (#13778150, ThermoFisher Scientific) according to the manufacturer’s protocol. *Dicer1* plasmid was kindly provided by Dr. Santisteban (IBMM, Madrid, Spain). Plasmid transfection was performed using Lipofectamine3000 reagent (L3000015, ThermoFisher Scientific) according to the manufacturer’s protocol.

### 4.5. Western Blotting

PBS-washed cells were lysed with LAEMMLI buffer supplemented with protease and phosphatase inhibitors and protein concentration was determined using Ionic Detergent Compatibility Reagent (IDCR) for Pierce (22663, ThermoFisher Scientific). Equal amounts of proteins (30 μg) were separated on a 6% polyacrylamide gel and transferred to a nitrocellulose membrane (or PVDF membrane for H2AX/pH2AX antibodies) which was incubated with the matching primary antibodies at 4 °C overnight and secondary HRP-conjugated antibodies at room temperature for 1 h: Dicer1 (A-2, sc-136981, SantaCruz, Dallas, TX, USA, 1/250); Actin (A2066, Merck, Rahway, NJ, USA, 1/1000), Vinculin (VIN-54, ab130007, Abcam, Cambridge, UK, 1/1000); Cyclin E (HE12, MA5-14336, ThermoFisher Scientific, 1/250); Cyclin D1 (DSC6, AB187364, Abcam 1/500); H2AX (07-627, Merck Milipore, 1/750); phospho-H2AX (p-ser139, NB100-384, Novus Biologicals, Centennial, CO, USA, 1/1000); Peroxidase AffiniPure Donkey Anti-Mouse IgG (H+L) (715-035-150, Jackson ImmunoResearch, West Grove, PA, USA); Peroxidase AffiniPure Donkey Anti-Rabbit IgG (H+L) (711-035-152, Jackson ImmunoResearch). Staurosporine-treated cells (10 μM, S5921, Sigma, St. Louis, MO, USA) were used as a positive control for H2AX/pH2AX. Proteins were visualized with Western Lightning Plus-enhanced chemiluminescence substrate (NEL103E001EA, Perkin Elmer, Waltham, MA, USA) and normalized to actin or vinculin. Quantification was performed using ImageJ 1.54d software.

### 4.6. Total RNA Sequencing and Analysis

Sequencing was performed at the BRIGHTcore facility (www.brightcore.be). RNA quality was checked via a Fragment analyzer (Agilent technologies). Indexed cDNA libraries were obtained using the TruSeq Stranded mRNA Sample Prep kit (20020595, Illumina) following manufacturer recommendations. The multiplexed libraries were loaded on a NovaSeq 6000 (Illumina) and sequences were produced using a 200 cycles reagent kit. Paired-end reads were mapped against the human reference genome GRCH38 using STAR_2.5.3a version 2.7 software to generate read alignments for each sample. Annotations Homo_sapiens.GRCh38.90.gtf were obtained from ftp.Ensembl.org. After transcripts assembling, gene level counts were obtained using HTSeq-0.9.1 and normalized to 20 million aligned reads. Data were computed on these values between the experimental conditions using the Degust and iDEP 1.1 (integrated Differential Expression and Pathway analysis) softwares. Genes differentially expressed were identified with an EdgeR quasi-likehood method.

### 4.7. Cell Proliferation Analysis

Four days after the transfection of *Dicer1* siRNA, cells were incubated with 5-ethynyl-2-deoxyuridine (EdU) for six hours at a final concentration of 10μM or bromodeoxyuridine (BrdU) for three hours at a final concentration of 10μM. Analysis was performed using The Click-iT™ Plus EdU Alexa Fluor™ 488 Flow Cytometry Assay Kit (C10633, ThermoFisher Scientific) or the APC BrdU Flow kit (552598, BD Biosciences, Aalst, Belgium). Cells were trypsinized and fixed according to the manufacturer’s protocol and subjected to flow cytometry analysis.

### 4.8. Apoptosis Analysis

Apoptosis levels were assessed four days post-transfection using the Caspase-Glo^®^ 3/7 Assay. Staurosporine-treated cells (10 μM, S5921, Sigma) were used as a positive control. Three days post-transfection, cells were counted and 6000 cells were seeded per well in a 96-well plate. Twenty hours after seeding, the Caspase-Glo^®^ 3/7 Assay System mixture was added, followed by a 60 min incubation. The supernatant was then transferred to a 96-well white-bottom plate, and luminescence was measured using a TECAN Infinite 200 pro luminometer. The blank control value was subtracted from each point. Each measure was an average of two duplicate wells. 

Additionally, apoptotic levels were determined using the Dead Cell Apoptosis Kit with Annexin V-FITC and Propidium Iodide (PI) for flow cytometry (V13242, ThermoFisher) following the manufacturer’s protocol. Staurosporine-treated cells (10 μM, S5921, Sigma) were used as a positive control. Cells were stained and analyzed using a flow cytometer to quantify the percentage of apoptotic cells.

### 4.9. Flow Cytometry Analysis

Flow cytometry was performed using a Fortessa from Becton, configured with four lasers (violet, blue, yellow-green, red). Data analysis was carried out using Diva software, version 9.

### 4.10. Invasion and Migration Analysis

Migration assays were performed using transwell migration chambers (24-well plate, 8.0 µm pore size, VWR 734-0038, Corning, Leuven, Belgium) and invasion assays were performed using transwell invasion chambers coated with matrigel (VWR 734-1047, Corning). Forty thousand cells were seeded into the upper compartment of the transwell chambers in media without FBS and the lower compartment was filled with media supplemented with 10% FBS 2% penicillin/streptomycin and 1% amphotericin. Twenty hours later, cells that had migrated or invaded were fixed, stained (26419, Polysciences Inc., Taipei, Taiwan), and counted.

### 4.11. Statistical Analyses

Statistical analyses were performed using Prism GraphPad 6.0. Data distribution was analyzed using the Shapiro–Wilk normality test. Statistically significant differences between two groups were determined using the Mann–Whitney *t*-test. Statistically significant differences for more than two groups were determined using the Kruskal–Wallis test. The columns represent the mean values and the error bars indicate the standard deviation (mean ± SD). All experiments were replicated at least three times independently. A *p*-value less than 0.05 was considered statistically significant. * *p* < 0.05, ** *p* < 0.01 and *** *p* < 0.001.

## 5. Conclusions

In conclusion, the regulation of *Dicer1* expression plays a critical role in thyroid tumorigenesis, and complete elimination or overexpression of *Dicer1* inhibits thyroid oncogenesis, revealing this protein as a promising therapeutic target for both diagnostic and prognostic applications for thyroid cancer. The therapeutic modulation of *Dicer1* and its associated proteins via molecules acting as activators, such as enoxacin, or inhibitors like CIB-3b, presents promising new approaches for cancer treatment [41,42], providing innovative ways to interfere with cancer progression and highlighting the importance of miRNA biogenesis in cancer progression.

## Figures and Tables

**Figure 1 ijms-25-10701-f001:**
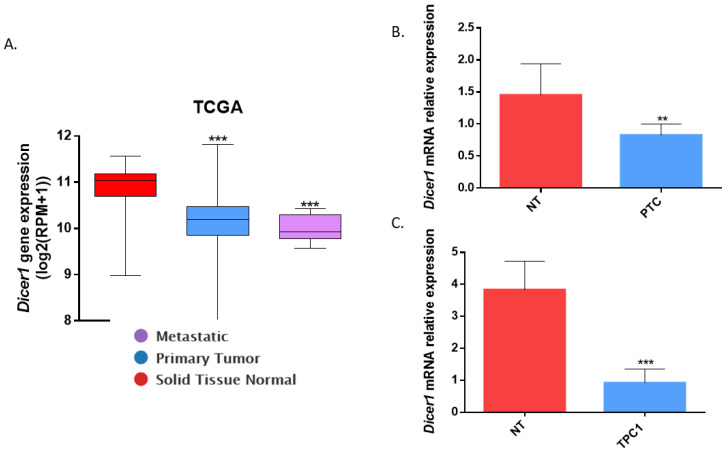
*Dicer1* mRNA expression was decreased in the human PTC and in the PTC-derived cell line TPC1. (**A**) Analysis of *Dicer1* mRNA expression in The Cancer Genome Atlas (TCGA) in normal samples (red, Solid Tissue Normal, n = 59), PTC (blue, primary tumors, n = 505), and metastases (purple, n = 8). The expression (log2 (RPM+1)) refers to the logarithm base 2 of the Reads Per Million (RPM) value increased by one. Statistically significant differences were determined using the Kruskal–Wallis test *** *p* < 0.001. (**B**) Analysis of *Dicer1* mRNA expression using RT-qPCR in 9 PTC (blue) and their adjacent normal tissue (red, NT). Statistically significant differences were determined using the Mann–Whitney test ** *p* < 0.01. (**C**) Analysis of *Dicer1* mRNA expression in the PTC-derived cell line TPC1 (blue, n = 7) compared with a pool of normal human thyroid tissues (red, NT, n = 8). The columns represent the mean values and the error bars indicate the standard deviation (mean ± SD). Statistically significant differences were determined using the Mann–Whitney test *** *p* < 0.001.

**Figure 2 ijms-25-10701-f002:**
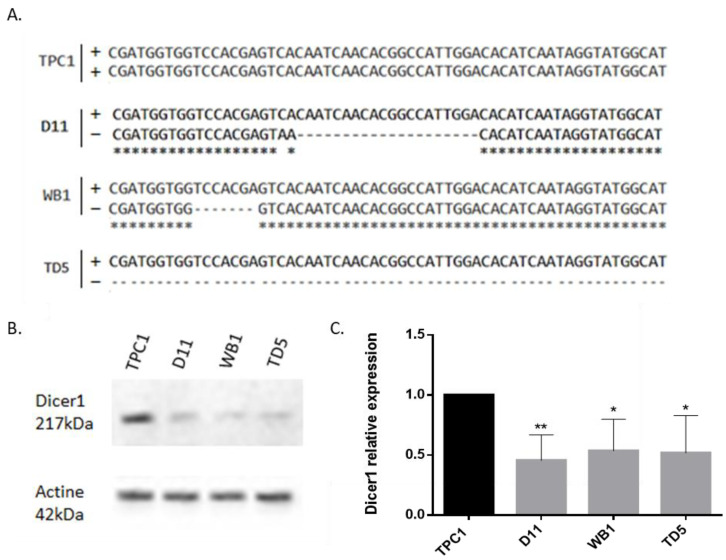
Generation of D11, WB1, and TD5 *Dicer1* (+/−) cell lines using CRISPR-Cas9. (**A**) DNA was extracted from isolated clones after transfection, amplified, cloned into a plasmid, and sequenced using Sanger sequencing. TPC1 presented two wild-type alleles (+); D11, WB1, and TD5 displayed a combination of one wild-type allele (+) and one knockout allele (−), with deletions of 20, 7, and 185 nucleotides, respectively. (**B**) Western blot analysis of Dicer1 protein expression in TPC1 and in CRISPR-Cas9 generated D11, WB1, and TD5 cell lines. The upper image corresponds to the detection of Dicer1 with a molecular weight of 217 kDa, while the lower panel corresponds to the detection of actin with a molecular weight of 42 kDa. (**C**) Western blot quantification of Dicer1 protein expression in TPC1 and in CRISPR-Cas9 generated D11, WB1, and TD5 cell lines (n = 8). Data were normalized to Dicer1 expression in TPC1. The columns represent the mean values, and the error bars indicate the standard deviation (mean ± SD). Statistically significant differences were determined using the Kruskal–Wallis test. * *p* < 0.05 and ** *p* < 0.01.

**Figure 3 ijms-25-10701-f003:**
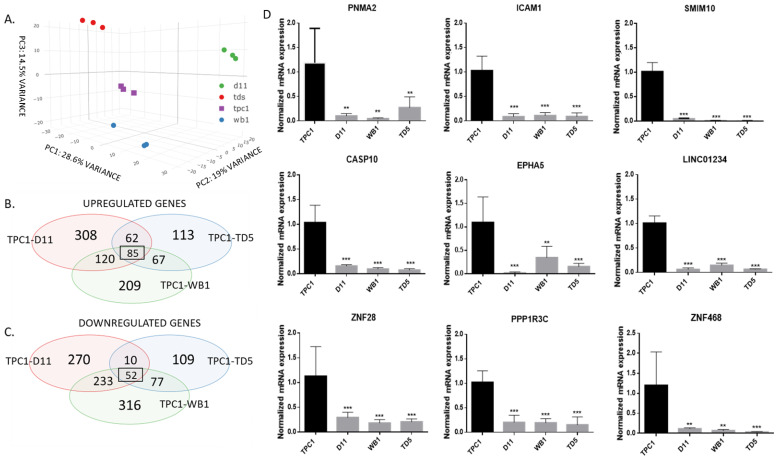
RNAseq analysis did not reveal major transcriptomic changes in heterozygous cell lines. (**A**) PCA analysis of gene variance between TPC1 (purple), D11 (green), WB1 (blue), and TD5 (red). (**B**,**C**) Analysis of common upregulated or downregulated genes between TPC1 and D11 (red), TPC1 and WB1 (green), and TPC1 and TD5 (blue). (**D**) PNMA2, ICAM1, SMIM10, CASP10, EPHA5, LINC01234, ZNF28, PPP1R3C, and ZNF468 mRNA expression analysis using RT-qPCR in TPC1, D11, WB1, and TD5 cell lines (n = 5). The columns represent the mean values and the error bars indicate the standard deviation (mean ± SD). Statistically significant differences were determined using the Kruskal–Wallis test. ** *p* < 0.01 and *** *p* < 0.001.

**Figure 4 ijms-25-10701-f004:**
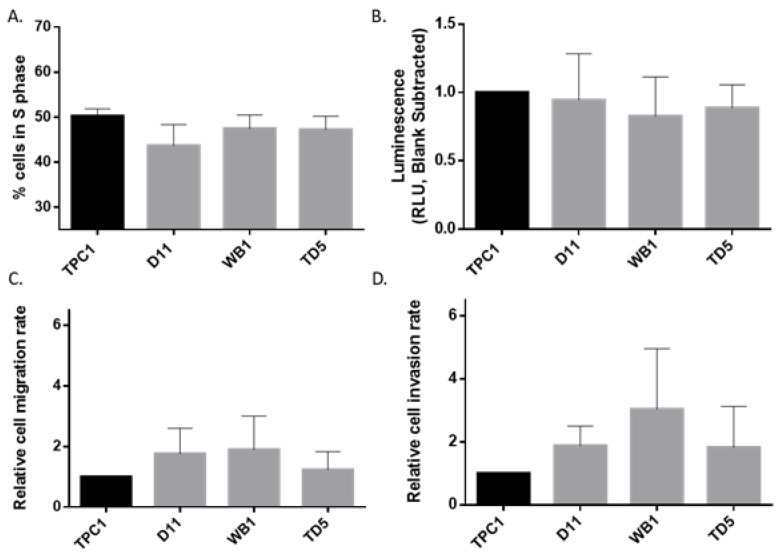
Heterozygous loss of Dicer1 did not impact proliferation, apoptosis, invasion or migration. (**A**) Cell proliferation was analyzed in TPC1, D11, WB1, and TD5 cell lines (n = 4). Cells were intracellularly stained with BrdU Staining Kit for Flow Cytometry. The percentage of BrdU positive cells (cells in S phase of the cell cycle) was determined via flow cytometry. (**B**) Apoptosis was analyzed using Caspase-Glo^®^ 3/7 Assay in TPC1, D11, WB1, and TD5 cell lines (n = 6). Each point represents the average of 2 wells. The blank control value has been subtracted from each point. (**C**) Migration analysis in TPC1 (n = 10), D11 (n = 6), WB1 (n = 10), and TD5 (n = 10) cell lines. (**D**) Invasion analysis in TPC1, D11, WB1, and TD5 cell lines (n = 5). The columns represent the mean values and the error bars indicate the standard deviation (mean ± SD). Data were normalized to TPC1. Statistically significant differences were determined using the Kruskal–Wallis test.

**Figure 5 ijms-25-10701-f005:**
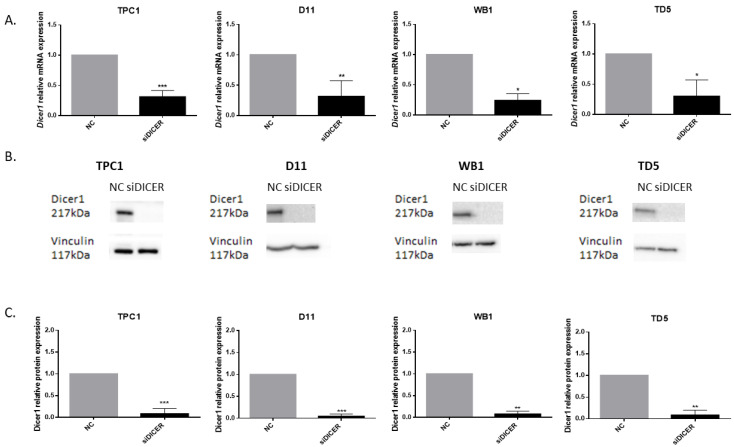
Dicer1 mRNA and protein expression were strongly reduced four days after Dicer1 siRNA transfection. (**A**) Dicer1 mRNA expression was analyzed using RT-qPCR in the TPC1 (n = 7), D11 (n = 5), WB1 (n = 4), and TD5 (n = 4) cell lines transfected with the negative control (NC) or with Dicer1 siRNA (siDICER). (**B**) Western blot analysis of Dicer1 protein expression in the same conditions. The upper image corresponds to the detection of Dicer1 with a molecular weight of 217 kDa, while the lower panel corresponds to the detection of vinculin with a molecular weight of 117 kDa. (**C**) Quantification of Dicer1 protein levels in TPC1 (n = 8), D11 (n = 7), WB1 (n = 6) and TD5 (n = 6) cells. Data were normalized to NC. The columns represent the mean values and the error bars indicate the standard deviation (mean ± SD). Statistically significant differences were determined using the Mann–Whitney test. * *p* < 0.05, ** *p* < 0.01 and *** *p* < 0.001.

**Figure 6 ijms-25-10701-f006:**
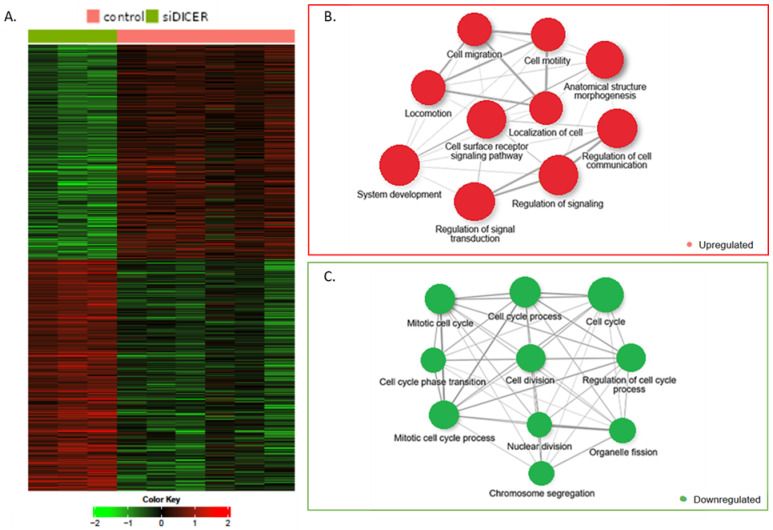
RNAseq analysis revealed transcriptomic alterations after transfection with Dicer1 siRNA in D11 cells. (**A**) Heatmap representation of the hierarchical clustering (Pearson < 0.0001) of genes selected with DESeq2 method, FDR 0.05, and fold change 1.5× (following Dicer1 siRNA (siDICER)) transfection vs. control (following transfection with negative control (NC)) (1123 upregulated genes, in red, and 1204 downregulated genes, in green). (**B**,**C**) Top 10 upregulated or downregulated pathways network obtained through the enrichment analysis of gene expression (FDR 0.05 and minimum fold change 1.5×) in D11 cells transfected with siDICER compared with controls using The Gene Ontology Biological Process database.

**Figure 7 ijms-25-10701-f007:**
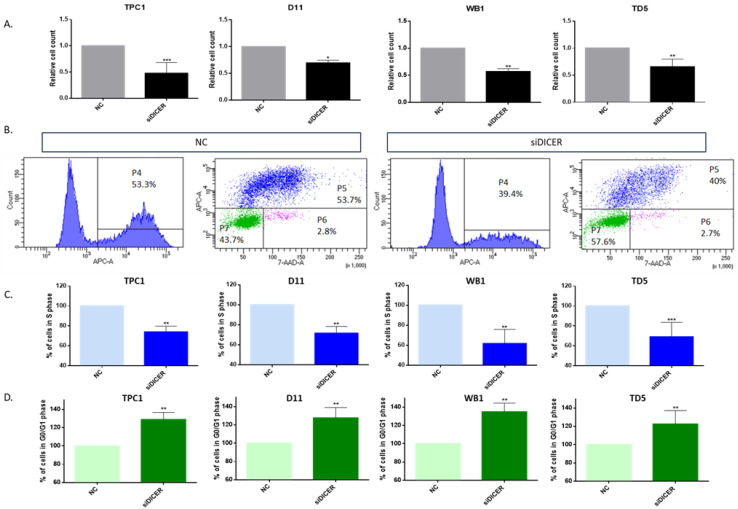
Total Dicer1 loss inhibited proliferation. (**A**) Cell counts were conducted in TPC1 (n = 7), D11 (n = 4), WB1 (n = 5), and TD5 (n = 5) cell lines transfected with the negative control (NC) or with Dicer1 siRNA (siDICER). (**B**) TPC1 cells transfected with the negative control (NC) or with Dicer1 siRNA (siDICER) were intracellularly stained with Anti-BrdU-APC-A and 7-AAD using the BrdU Staining Kit for Flow Cytometry. For each condition (NC or siDICER), the left panel shows only BrdU positive cells (S phase, P4). On the right panel, BrdU positive cells are stained in blue (S phase, P5) and BrdU negative cells are stained in green (low 7-AAD concentrations, G0/G1 phases, P7) or purple (high 7-AAD concentrations, G2/M phases, P6). (**C**) Quantification of cells in P5 (S phase) in TPC1 (n = 7), D11 (n = 4), WB1 (n = 5), and TD5 (n = 5) cell lines following negative control (NC) or Dicer1 siRNA (siDICER) transfection. (**D**) Quantification of cells in P7 (G0/G1 phases) in the same experimental conditions. All experiments were conducted four days after transfection. Data were normalized to the NC. The columns represent the mean values and the error bars indicate the standard deviation (mean ± SD). Statistically significant differences were determined using the Mann–Whitney *t*-test. * *p* < 0.05, ** *p* < 0.01 and *** *p* < 0.001.

**Figure 8 ijms-25-10701-f008:**
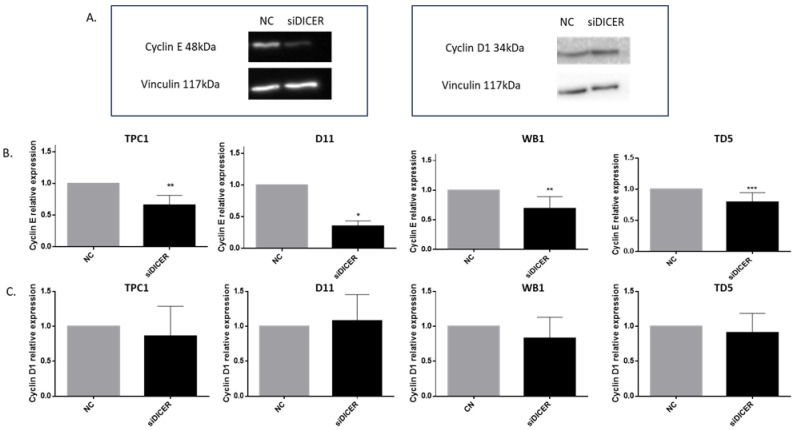
Cyclin E but not cyclin D1 was downregulated following Dicer1 siRNA transfection. (**A**) Western blot analysis of cyclin E and cyclin D1 protein expression in TPC1 cells transfected with the negative control (NC) or with Dicer1 siRNA (siDICER). The upper image corresponds to the detection of cyclin E or D1 (48 kDa and 34 kDa, respectively) while the lower panel corresponds to the detection of vinculin (117 kDa). (**B**) Western blot quantification of cyclin E expression in TPC1 (n = 6), D11 (n = 4), WB1 (n = 6), and TD5 (n = 7) cell lines following negative control (NC) or Dicer1 siRNA (siDICER) transfection. (**C**) Western blot quantification of cyclin D1 expression in the same experimental conditions (n = 6). Cyclin expression was normalized to vinculin expression and data were normalized to the NC. The columns represent the mean values and the error bars indicate the standard deviation (mean ± SD). Statistically significant differences were determined using the Mann–Whitney *t*-test. * *p* < 0.05, ** *p* < 0.01 and *** *p* < 0.001.

**Figure 9 ijms-25-10701-f009:**
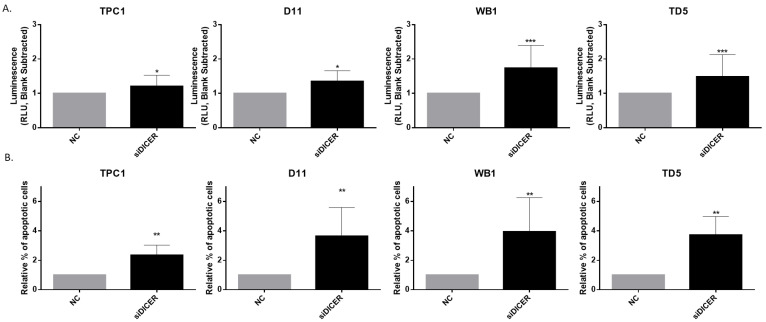
Total loss of Dicer1 resulted in an increase in apoptosis. (**A**) Apoptosis was analyzed using Caspase-Glo^®^ 3/7 Assay in TPC1 (n = 10), D11 (n = 7), WB1 (n = 12), and TD5 (n = 12) cells transfected with the negative control (NC) or with Dicer1 siRNA (siDICER). (**B**) The percentage of apoptotic cells was determined using a Dead cell apoptosis kit with Annexin V FITC and Propidium Iodide for flow cytometry in TPC1 (n = 5), D11 (n = 6), WB1 (n = 6), and TD5 (n = 6) cells transfected with the negative control (NC) or with Dicer1 siRNA (siDICER). Data were normalized to NC. The columns represent the mean values and the error bars indicate the standard deviation (mean ± SD). Statistically significant differences were determined using THE Mann–Whitney *t*-test. * *p* < 0.05, ** *p* < 0.01 and *** *p* < 0.001.

**Figure 10 ijms-25-10701-f010:**
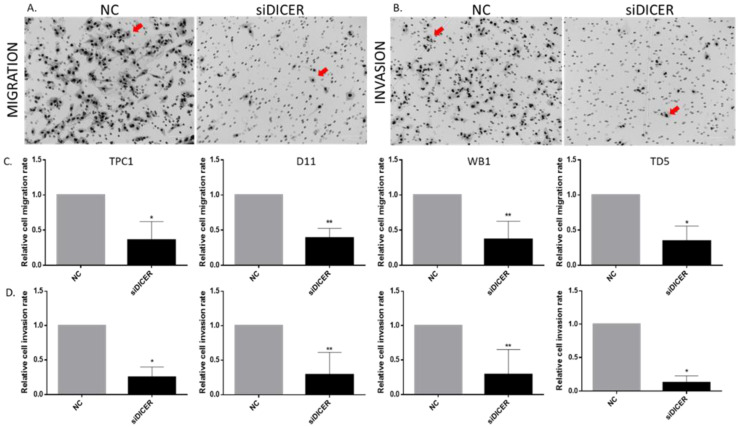
Dicer1 total loss inhibited migration and invasion. The number of cells migrating (**A**) or invading (**B**) was counted 4 days after transfection with the negative control (NC) or Dicer1 siRNA (siDICER). The red arrows point to examples of the cells. (**C**) Quantification of migrating cells in TPC1 (n = 4), D11 (n = 5), WB1 (n = 5), and TD5 (n = 4) cell lines. (**D**) Quantification of invasive cells in TPC1 (n = 4), D11 (n = 5), WB1 (n = 5), and TD5 (n = 4) cell lines. Data were normalized to NC. The columns represent the mean values and the error bars indicate the standard deviation (mean ± SD). Statistically significant differences were determined using the Mann–Whitney *t*-test. * *p* < 0.05 and ** *p* < 0.01.

**Figure 11 ijms-25-10701-f011:**
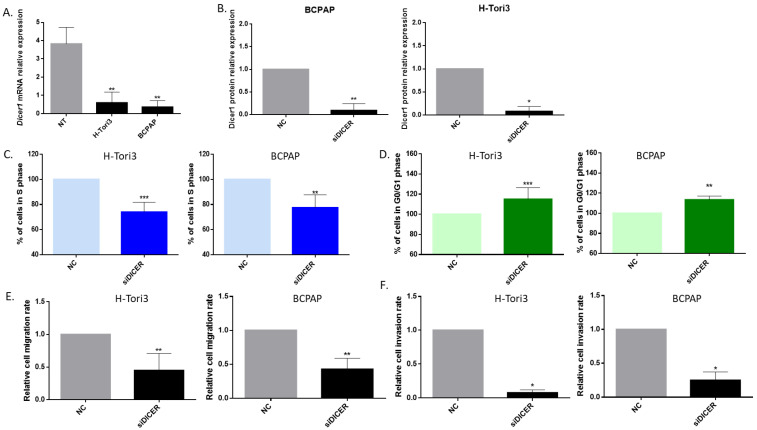
Dicer1 knockdown inhibited proliferation as well as migration and invasion in BCPAP and H-Tori3 cells. (**A**) Analysis of Dicer1 mRNA expression in BCPAP (n = 4) and H-Tori3 (n = 7) cells compared with a pool of normal human thyroid tissues (NT, n = 8). (**B**) Dicer1 protein expression levels in BCPAP (n = 6) and H-Tori3 (n = 4) cells following negative control (NC) or Dicer1 siRNA transfection (siDICER). (**C**) Quantification of cells in S phase in H-Tori3 (n = 8) and BCPAP (n = 5) cells. (**D**) Quantification of cells in G0/G1 phase in H-Tori3 (n = 8) and BCPAP (n = 5). (**E**) Quantification of migrating cells in H-Tori3 (n = 6) and BCPAP (n = 6). (**F**) Quantification of invasive cells in H-Tori3 (n = 4) and BCPAP (n = 4). All experiments were conducted four days after transfection with the negative control (NC) or with Dicer1 siRNA (siDICER). Data were normalized to the NC. The columns represent the mean values and the error bars indicate the standard deviation (mean ± SD). Statistically significant differences were determined using the Mann–Whitney *t*-test. * *p* < 0.05, ** *p* < 0.01 and *** *p* < 0.001.

**Figure 12 ijms-25-10701-f012:**
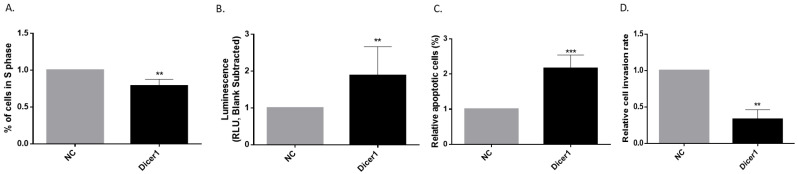
Increasing Dicer1 expression inhibited proliferation and invasion while raising apoptosis. All experiments were performed in TPC1 cells transfected with Dicer1 plasmid (Dicer1) 48 h after transfection. (**A**) Quantification of cells in S phase (n = 6). (**B**) Apoptosis was analyzed using Caspase-Glo^®^ 3/7 Assay (n = 9). (**C**) The percentage of apoptotic cells was determined using a Dead cell apoptosis kit with Annexin V FITC and Propidium Iodide for flow cytometry (n = 9). (**D**) Quantification of invasive cells (n = 6). Data were normalized to NC (negative control: cells incubated in the presence of lipofectamine 3000). The columns represent the mean values and the error bars indicate the standard deviation (mean ± SD). Statistically significant differences were determined using the Mann–Whitney *t*-test. ** *p* < 0.01, *** *p* < 0.001.

**Figure 13 ijms-25-10701-f013:**
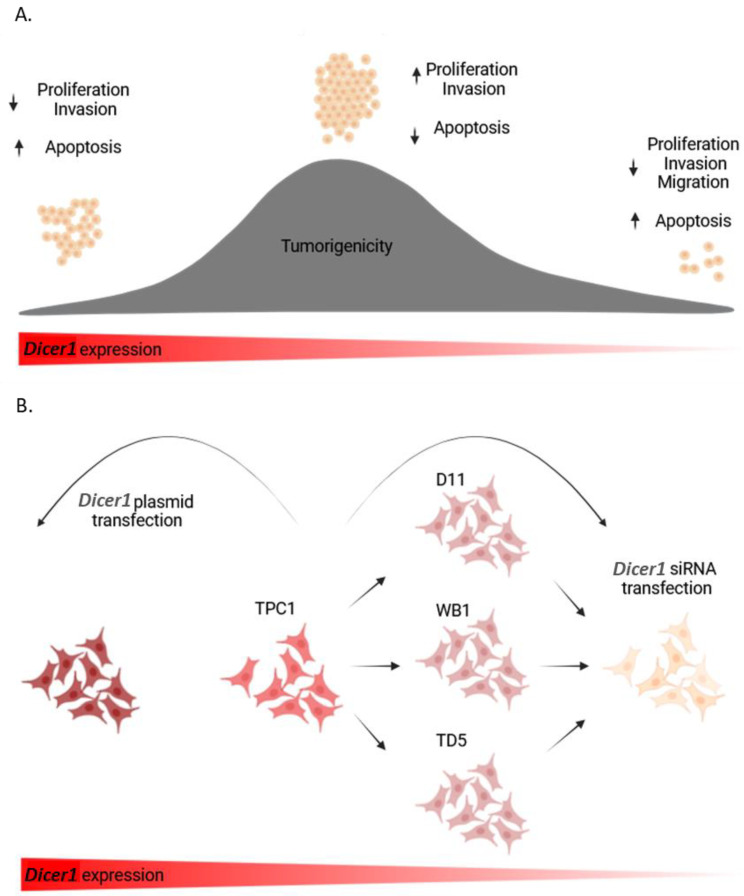
*Dicer1* exerted dosage-dependent effects in thyroid cells. (**A**) Very low levels or total absence of *Dicer1* resulted in reduced (↓) proliferation, invasion and migration, along with increased (↑) apoptosis, suggesting an inhibition of tumorigenesis. Similarly, high levels of *Dicer1* led to comparable proliferation, invasion, and apoptosis outcomes. Intermediate levels of *Dicer1* (as in TPC1, D11, WB1, and TD5 cells) led to increased proliferation and reduced apoptosis, potentially promoting tumorigenesis. (**B**) High levels of *Dicer1* were obtained via plasmid transfection. TPC1 had intermediate levels of *Dicer1* and heterozygous *Dicer1* (+/−) D11, WB1, and TD5 cells showed a 50% decrease in *Dicer1* expression compared with TPC1 cells. Following *Dicer1* siRNA transfection, *Dicer1* was almost completely lost.

## Data Availability

The original data presented in the study are openly available in the Gene Expression Omnibus (GSE273204 and GSE273129).

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
