# Peer review of "Understanding the Dosage-Dependent Role of Dicer1 in Thyroid Tumorigenesis"

_ijms, 2024, doi:10.3390/ijms251910701_

Round 1

Reviewer 1 Report

Comments and Suggestions for Authors

In the manuscript, the authors describe the role of Dicer1 as possible treatment target in thyroid cancer. Following Dicer1 siRNA transfection, in vitro experiments revealed changes in proliferation, cell cycle, and motility. As a result, the cell cycle slowed down and the number of invasive and migratory cells dropped. Furthermore, transfection with the Dicer1 plasmid increased apoptosis and decreased invasion and proliferation. The article is well written, but major issues should be addressed to improve it.

1. The authors should consider including recent articles regarding miRNAs in the introduction: DOI: 10.3390/biomedicines12030658, DOI: 10.3390/ijms25137448

2. The authors should consider inserting the protocol number approved by the ethics committee in the methods.

3. Furthermore, the description of the in silico study performed is completely missing. The authors should include it in the methods, defining whether any tools were queried and the date of access to them.

4. In the results to improve them, the authors should also include a wound healing assay as a migration assay while to define cell death they should perform a test with annexin/pi

Reviewer 2 Report

Comments and Suggestions for Authors

The manuscript by Rojo-Pardillo and colleagues foccued in better understand the response of thyroid cancer cells of Dicer1 in vitro due to the fact that the loss of one allele promotes tumorigenesis while the complete loss of Dicer1 prevents tumor formation. The results showed reduced proliferation and invasion as well as increased apoptosis. The authors hypothesized that Dicer1 as an attractive target for novel therapeutic strategies for thyroid cancer, a frequent endocrine tumors with an increasing incidence. 

Comments

1. Abstract Section

The authors should provide a more comprehensive discussion of the significance of studying thyroid cancer.

2. Introduction Section

The authors should provide a more detailed explanation of the thyroid cancer, including their prevalence, characteristics, and potential treatment options. Additionally, they should elaborate on the specific reasons why studying thyroid cancer is crucial, such as its impact on patient health, the need for improved treatment strategies, and its potential implications for understanding other types of cancer.

3. Results Section

a. While Caspase-Glo assays are a valuable tool for studying apoptosis, it's recommended to use them in conjunction with other methods to obtain a comprehensive understanding of apoptotic processes.

b. The authors should provide a detailed explanation of the rationale behind selecting specific time points for each experimental condition.

c. Given the unexpected elevation in migration rates observed in the results and the authors' inability to provide a satisfactory explanation, additional assays should be conducted to elucidate the underlying mechanisms.

d. The authors should provide a detailed description of the histone H2AX phosphorylation analysis, including the specific methods used, the experimental conditions, and the controls employed. This would strengthen the validity and reproducibility of the study's findings.

e. The HGNC endorses the use of italics to denote genes and mRNA: Please describe all gene and mRNA in italic format.

4. Methods Section

a. Ethics committee approval: Please provide provide an ID and date of issue of the local ethics committee approval for this study.

b. Flow cytometry section: Please add a subsection containing all flow assay and analysis in the Methods Section.

c. Statistical analysis: Please add a subsection containing all statistical analysis in the Methods Section.

d. Please provide the size of the transwell migration chambers. 

Comments on the Quality of English Language

Minor editing of English language required
